# Do Androgenic Pattern, Insulin State and Growth Hormone Affect Cardiorespiratory Fitness and Strength in Young Women with PCOS?

**DOI:** 10.3390/biomedicines10092176

**Published:** 2022-09-02

**Authors:** Veronica Baioccato, Giulia Quinto, Sara Rovai, Francesca Conte, Francesca Dassie, Daniel Neunhäeuserer, Marco Vecchiato, Stefano Palermi, Andrea Gasperetti, Valentina Bullo, Valentina Camozzi, Roberto Vettor, Andrea Ermolao, Roberto Mioni

**Affiliations:** 1Sport and Exercise Medicine Division, Department of Medicine, Padova University Hospital, Regional Centre for Exercise Prescription in Chronic Diseases, 35128 Veneto, Italy; 2Department of Medicine, Clinica Medica 3, Azienda Ospedaliera Padova, University of Padova, 35122 Padova, Italy; 3Public Health Department, University of Naples Federico II, 80138 Naples, Italy; 4Endocrinology Division, Department of Medicine, Padova University Hospital, 35128 Padova, Italy

**Keywords:** polycystic ovary syndrome, cardiopulmonary exercise test, functional capacity, muscular strength, exercise prescription

## Abstract

In this study, cardiorespiratory fitness (CRF) and strength level were assessed in women with and without polycystic ovary syndrome (PCOS), matched for age, body composition, androgenic pattern and insulinemic pattern. Patients with and without PCOS were evaluated at the Endocrinology Unit and Sport Medicine Division to assess endocrinological (insulinemic, androgenic pattern and growth hormone), anthropometric (with DEXA) and functional parameters (with cardiopulmonary exercise test and handgrip test), as well as physical activity level (with the Global Physical Activity Questionnaire). A total of 31 patients with PCOS and 13 controls were included. No statistically significant differences were found between groups in terms of age, body mass index, body composition, androgenic pattern, insulin state, growth hormone and physical activity level. The PCOS group demonstrated significantly better cardiorespiratory fitness (VO_2_max per kg (30.9 ± 7.6 vs. 24.8 ± 4.1 mL/kg/min; *p* = 0.010), VO_2_max per kg of fat-free mass (52.4 ± 8.9 vs. 45.3 ± 6.2 mL/kg/min; *p* = 0.018)), strength levels (handgrip per kg (0.36 ± 0.09 vs. 0.30 ± 0.08; *p* = 0.009), handgrip per kg of fat-free mass (13.03 ± 2.32 vs. 11.50 ± 1.91; *p* = 0.001)) and exercise capacity (METs at test (14.4 ± 2.72 vs. 12.5 ± 1.72 METs; *p* = 0.019)). In this study, women with PCOS showed a better cardiorespiratory fitness and strength than the control group. The only determinant that could explain the differences observed seems to be the presence of the syndrome itself. These results suggest that PCOS per se does not limit exercise capacity and does not exclude good functional capacity.

## 1. Introduction

Polycystic ovary syndrome (PCOS) is the most common endocrine pathology affecting reproductive-aged women, with a prevalence ranging between 8% and 13%, and it is characterised by chronic anovulation, menstrual dysfunction and hyperandrogenism [1]. Frequently, PCOS is associated with several metabolic risk factors, such as obesity, insulin resistance and hyperinsulinemia, hypertension, dyslipidaemia and type 2 diabetes and it has a negative impact on quality of life [2,3]. Exercise is the first-line therapy for PCOS, especially if it is tailored to a patient’s characteristics, as evaluated by functional and cardiopulmonary exercise tests [4]. Indeed, it is well known that cardiorespiratory fitness (CRF), measured as maximal aerobic capacity (VO_2_max), is a strong independent predictor of cardiovascular and all-cause mortality in men and women [5]. Moreover, VO_2_max is a primary determinant of one’s ability to sustain physical activity, and it has an impact on quality of life, cardiovascular health and engagement in exercise training [6,7]. The literature offers contrasting reports regarding VO_2_max in the PCOS population. Some studies agree that this parameter is impaired with respect to age and body mass index (BMI)-matched healthy controls [8,9,10]. Other authors describe a normal CRF in women with PCOS, showing no differences in VO_2_max between overweight women with and without PCOS, who were matched in terms of age and BMI [11,12]. This topic is currently debated in the literature, perhaps because the PCOS population is variable, depending on several phenotypes. This heterogeneity in anthropometric and hormonal patterns may justify the variability in the results discussed above.

A complete functional assessment also includes a strength measurement, as measured via the handgrip test [13]. It is well known that improving both cardiorespiratory fitness and muscle strength may be the most effective behavioural strategy with which to reduce the risk of metabolic syndrome and all-cause mortality [14,15]. The underlying physiological mechanisms between strength and cardiovascular benefits are not fully understood, particularly in the PCOS population [16]. Conflicting results were found in the literature when assessing strength in women with PCOS [17,18]. Thus, the purpose of our observational study is to assess CRF and strength in women with and without PCOS who were matched in terms of age, body composition and androgenic and insulinemic patterns. 

## 2. Materials and Methods

We included 44 Caucasian patients (aged 18–46 years old) who underwent a clinical evaluation for endocrinological disorders at the Endocrine Unit of Internal Medicine 3, Padova University Hospital; subjects did not present any contraindications to performing a cardiopulmonary exercise test (CPET). Thirty-one of these women were affected by PCOS, according to the Rotterdam criteria [19]. The PCOS subjects were also classified according to the phenotype described by the Androgen Excess–PCOS Society [20]. In particular, 20 of these 31 women (64.5%) had the classic PCOS phenotype (type 1: i.e., they showed both clinical and/or biochemical hyperandrogenism, chronic oligo-anovulation and polycystic ovary morphology), whereas 11 (34.6%) had either the ovulatory (type 2: 25.0%) or normoandrogenic (type 4: 9.6%) phenotypes. The remaining 13 women, who were being evaluated for functional disorders but were not affected by PCOS, were considered a control group. Hyperandrogenism was defined by the clinical presence of hirsutism, based on a Ferriman–Gallwey modified score [21,22]. Patients were assessed using a transvaginal ultrasound (US Philips, Probe of 8–10 MHz, Milan, Italy) to evaluate ovary structure and volume. At the time of the study, no patients showed any systemic and/or endocrine diseases, such as hyperprolactinemia, pituitary or thyroid diseases, Cushing’s syndrome and acanthosis nigricans or had a personal or family history of type 2 diabetes mellitus or adrenal enzyme deficiency. Any medication or oral contraceptives had been suspended for at least 3 months before the evaluation. All the participants underwent a three-hour oral glucose tolerance test (OGTT), maintaining an intake of at least 300 g of glucose in the two days before the test [23]. Normal weight was considered a BMI ranging between 18.0 and 24.9 (kg/m^2^) or obese at a BMI higher than 30, as well as being considered normoinsulinemic (maximum insulin peak ≤ 60 mIU/L, ISI ≥ 6, and HOMA-ir < 1.8) or hyperinsulinemic (maximum insulin peak > 70 mIU/L, ISI < 4, and HOMA-ir ≥ 2.5) [24]. All subjects were non-smokers and had a low–moderate physical activity level (>500 METs/min/week); none drank alcoholic beverages. An assessment of menstrual history, recording menses in the 12-month period before and during the study, was carried out for each subject. The investigational nature of the study was explained to all participants, and informed consent was obtained after an interview, in which the study protocol was explained in detail. The study was approved by the Ethics Committee of the Padova University Hospital (n. 0608). 

All blood samples were obtained during the early follicular phase (days 2–5) of a spontaneous or progesterone-induced menstrual cycle. Glucose was measured by means of the glucose oxidase method (Gluco-quant^®^; Roche Diagnostics GmbH, Mannheim, Germany). LH, FSH and estradiol were assessed by a competitive immunoassay with the use of an electrochemiluminescence immunoassay (ECLIA; Roche Diagnostics GmbH, Mannheim, Germany). GH was measured by chemiluminescent immunometric assay (Medical System S.p.a., Genova, Italy) [25]. Serum Testosterone and DHEA-s were further measured by competitive immunometric chemiluminescent enzyme immunoassay (Immulite 2000; Diagnostic Products Corporation, Los Angeles, California, USA). Androstenedione and 17-OH-P were measured by enzyme immunoassay (ELISA) (DRG Instruments GmbH, Marburg, Germany). SHBG was measured by two-site chemiluminescent immunometric assay (Immulite 2000; Diagnostic Products Corporation, Los Angeles, USA). Cholesterol and triglycerides were determined by enzymatic methods (Hitachi automatic analyser 717). All patients were also evaluated at the Sport and Exercise Medicine Division. Clinicians and kinesiologists performing the functional evaluations were blinded to the group allocation. In addition, CPET was performed on a treadmill (COSMOS, T170 DE-med model), with gas analysis (Masterscreen CPX Jaeger, Carefusion, Hoechberg, Germany) and electrocardiogram monitoring during the entire test (Cardiosoft EKG System, GE, USA). Blood pressure was measured by a trained medical doctor using the auscultatory method. Breath-by-breath ventilation was measured with a low-resistance turbine to determine inspiratory and expiratory volumes and flow during the exercise test. Inspired and expired gases were sampled continuously at the mouth and analysed for concentrations of O_2_ and CO_2_ via mass spectrometry after calibration with analysed gas mixtures. The treadmill test consisted of a maximal Bruce Ramp protocol, characterised by progressive speed and inclination increments. The interruption criterion was the achievement of a Rate of Perceived Exertion ≥ of 18/20 on the Borg Scale [26], which is associated with a heart rate (HR) value ≥ 85% of the age-predicted maximum HR (220 bpm-age) and/or with a respiratory exchange ratio (RER) value > 1.10. A treadmill was preferred to a cycle-ergometer because of its more reliable value for VO_2max_, which is not limited by peripheral muscle fatigue [27,28]. Body composition was evaluated via Dual X-ray Absorptiometry (DXA) (model DXA QDR 4500 W, software Version 12.6; Hologic, Bedford, MA, USA). This method followed the International Society for Clinical Densitometry guidelines [29]. Handgrip dynamometer was used to evaluate the dominant and non-dominant upper limb strength (Baseline, Elmsford, NY, USA). The fit of the hand on the grip was adjusted to accommodate the right size of the participant. The elbow was flexed to 90° to guarantee the strongest grip strength measurement [30]. The results were evaluated in percentile form with respect to the reference population and as raw data [13]. The physical activity level was assessed with the Global Physical Activity Questionnaire (GPAQ). Women were classified either as “active” or “sedentary” following the World Health Organization (WHO) recommendations for weekly levels of physical activity [31]. A validated Italian version of the questionnaire was administered by a trained kinesiologist to ensure data reliability [32].

Data were entered into a spreadsheet and analysed with IBM SPSS Statistics (IBM Corp. Released 2017. IBM SPSS Statistics for Windows, Version 25.0., IBM Corp., Armonk, NY, USA). The prevalence of insulin sensitivity was evaluated with Pearson’s chi-square test. All other variables were analysed with a Shapiro–Wilk test to assess the normality of the distributions. Two-tailed Student *t*-tests for independent samples and non-parametric tests were performed to compare the PCOS and non-PCOS groups, respectively, to determine the normality of the data. Linear correlations were evaluated with Pearson or Spearman’s rho correlation coefficient. Statistical significance was set at *p* ≤ 0.05.

## 3. Results

Thirty-one patients with PCOS and thirteen controls were included. The baseline characteristics and hormonal parameters of the study population are represented in Table 1, considering patients with or without PCOS for inter-group comparison. 

No statistically significant differences were found between groups in terms of age, BMI, body composition, physical activity level, androgenic pattern, insulin state [33] and growth hormone (GH) concentrations. Functional characteristics, such as CRF, exercise capacity and muscle strength data, are described in Table 2. 

The PCOS patients showed a significantly higher VO_2_max per kg of body weight (VO_2_maxkg), VO_2_max per kg of fat-free mass (VO_2_maxFFM), oxygen uptake efficiency slope (OUES), handgrip per kg (H_Kg_), handgrip per kg of fat-free mass (H_FFM_) and exercise capacity (Figure 1). 

The CRF values, expressed as the absolute value of VO_2_max and handgrip, showed a similar trend, although it did not reach statistical significance. Considering the entire population, VO_2_maxFFM values were positively correlated with exercise capacity (r = 0.698, *p* < 0.0001) and PCOS per se (r = 0.362, *p* = 0.02), while no correlations were observed with physical activity level or androgenic pattern. Additionally, VO_2_maxFFM was inversely correlated with age (r= −0.396, *p* = 0.049) and BMI (r = −0.560, *p* < 0.001). H_FFM_ was only positively correlated with PCOS per se (r = 0.356, *p* = 0.024), and no correlation was found with physical activity level, androgenic pattern or age; in contrast, H_FFM_ was linearly and inversely correlated with BMI (r = −0.486, *p* = 0.001). As summarised in Figure 2 and Figure 3, when analysing PCOS subjects separately, VO_2_maxFFM did not correlate with insulinemic parameters. Insulin peak during the stimulation test did not correlate with VO_2_maxFFM or VO_2_maxkg, while it was inversely associated with H_FFM_ (r = −0.412, *p* = 0.037).

## 4. Discussion

This study suggests that PCOS patients with obesity have greater CRF, functional capacity and handgrip strength compared to matched controls.

The VO_2_max value considered the gold standard with which to determine a subject’s maximal aerobic capacity and—when expressed as VO_2_maxFFM—it can help explain differences in VO_2_maxkg in patients with PCOS with respect to a control population, as suggested by Giallauria et al. [10]. To our knowledge, this is the first study reporting that CRF, based on VO_2_maxFFM, VO_2_maxkg and OUES, is higher in PCOS women with obesity than a matched control group. Exercise capacity, i.e., the maximum amount of physical exertion that a subject can sustain, was greater in the PCOS group as well. While, in some studies, these parameters were not analysed [34,35], other studies found conflicting results [36]. Our findings support the idea that functional efficiency is not compromised in PCOS patients; indeed, these patients have globally higher CRF values.

This is a very debated topic in the literature. In fact, Cosar et al. observed comparable CRF findings between overweight and sedentary PCOS women vs. matched controls when tested on a bike ergometer [11]. Again, similar results were found by Thomson et al.’s study, in which patients with obesity with and without PCOS were tested on a treadmill [37]. On the other hand, several studies have reported a lower CRF in PCOS patients [9,38,39], although their controls were not always comparable in terms of androgenic pattern and/or insulin resistance [8]. Moreover, maximal aerobic capacity corrected for individual fat-free mass (VO_2_maxFFM), when evaluated via bio-impedance analysis, was lower in women with PCOS than controls [9]. However, it is well known that this method is not highly reliable in evaluating body composition in subjects with obesity and overweight [40]; for these reasons, we assessed FFM by using DEXA, which is currently considered the gold standard.

Because VO_2_maxFFM may be influenced by several factors, we attempted to consider all of them in interpreting our results. Initially, in agreement with other studies [8,10], we observed an inverse correlation between insulin resistance and CRF, but only when this was expressed as VO_2_maxkg, not as VO_2_maxFFM (r =−0.44; *p =* 0.012 and r =−0.267; *p =* 0.162, respectively). Our results confirm that insulin resistance is related to reduced VO_2_maxkg in both overweight and obese women affected by PCOS, but this link is suppressed when only fat-free mass is considered. Our data seem to support the hypothesis that insulin resistance plays a major role in impairing CRF based on several mechanisms, including impaired mitochondrial function, reduced insulin-induced endothelial nitric oxide synthase activity [41], increased autonomic sympathetic tone [42], decreased vagal tone and impaired baro-reflex activity [43]. On the other hand, higher VO_2_maxFFM values could be explained by the absence of insulin resistance in both groups [8].

Another reason for a better VO_2_maxFFM in the PCOS group could be linked to androgenic pattern. Indeed, in the literature, hyperandrogenism is related to worse CRF, especially in PCOS women, as described by other authors [39]. However, in this investigation, PCOS women and controls had comparable androgenic patterns; thus, it is reasonable to affirm that androgenic levels cannot explain the different CRF values observed in our population. It is well known that low and moderate physical activity levels can decrease androgen levels. This could be due to a reduction in adipose tissue and/or insulin resistance, which increase SHBG levels, with a consequent reduction in free androgen levels [44,45]. However, in our patients, we can rule out this mechanism because they presented comparable physical activity levels, as well as similar androgenic and insulinemic patterns.

Muscle strength is a marker of exercise tolerance [14], and its impairment is frequently associated with disability in women with obesity [46]. Moreover, this physical domain has a strong impact on quality of life and exercise capacity. In this study, we assessed strength with handgrip, which is frequently used as a prognostic marker of mortality and clinical index in patients, as well as a general predictor of total body strength [13,14]. Similar to the VO_2_maxFFM data, the HG data were also scaled based on the dominant arm’s fat-free mass, obtaining handgrip fat-free mass (H_FFM_), a datapoint amended by other potential confounding factors. In our study, patients with PCOS had higher H_FFM_ values compared to the controls, although the causes are not yet clear. Kogure et al. found that women with PCOS had greater muscle strength with respect to controls matched for body composition, but they justified this result based on the difference in androgenic patterns present in their groups [17]. Another study comparing PCOS subjects with hyperandrogenism and controls showed the same strength levels [18]. Considering the presence of a pathophysiological link, a potential influence on the part of hyperinsulinism and/or IR on strength levels cannot be excluded in PCOS subjects [47,48]. Unfortunately, previous investigations did not explore this relationship [18,49]. In our study, the lack of differences in androgenic pattern and insulinemic state between groups seems to exclude their role as determinants of strength levels.

Growth hormone and physical activity levels may influence both CRF and strength. It is well known that GH increases muscle mass and strength, and some studies reported higher GH levels in PCOS subjects [50,51]. However, the similar GH plasma levels between our PCOS patients and controls, all of which fell within the normal range, and the lack of difference in body fat-free mass seem to exclude any influence on strength levels [52].

The impact of physical activity level on CRF and strength is debated too, even if, in the literature, there is not a standardised method of quantification. In some studies, these data were quantified as leisure time physical activity [9,10], while others did not take this data into account [8]. Several studies considered only sedentary subjects, probably to avoid any confounding factors affecting hormonal, cardiopulmonary and strength parameters [11,35,37]. To better quantify physical activity level and limit its confounding effect, we administered the GPAQ questionnaire. According to this tool, both groups were engaged in low-to-moderate activity. All had similar levels of activity and reached the minimum recommended amount of 150 min/week of physical activity [31].

In our study, the only determinant that could explain the differences observed between PCOS women and controls seems to be the presence of the syndrome itself. Thus, PCOS per se could represent an intrinsic predisposition to better strength and CRF. This concept is already debated in the literature, and some hypotheses have been put forward, such as congenital virilisation birth [53], maternal PCOS, low birth weight related to disturbed fetal nutrition [54] and the enhancement of GABAergic innervation, which modifies LH secretion in GnRH-induced PCOS [55].

It is important to note that our study protocol did not include an analysis of muscle fibre composition [56], which could have been useful in explaining the differences in strength level and aerobic capacity. In fact, in the future, it may be helpful to add a magnetic resonance exam to characterise muscle composition.

The strength of this study is the matching groups, which had the same age, body composition, physical activity level, androgenic pattern and absence of insulin resistance. However, there are some limitations as well. The group sizes were small, and the subjects analysed were obese, probably due to the random sample selection. Furthermore, the role of hyperinsulinism cannot truly be evaluated, because PCOS per se represents a natural predisposition to an impaired insulin pattern [57].

## 5. Conclusions

In conclusion, our data showed that obese subjects with PCOS have better cardiorespiratory fitness and strength than controls, and this finding seems to be related with PCOS per se. These results could suggest that PCOS per se does not limit exercise capacity, but it is a clinical condition compatible with good functional capacity when compared to the healthy general population too. However, the lack of agreement with the findings of previous studies [38] suggests the need for further investigations on this topic.

Finally, physical activity is considered a first-line therapy for this syndrome in order to improve anthropometric, cardiovascular and metabolic profiles [58]. This should include both active behaviour and structured exercise programs, which seem to yield major health benefits [7,58,59,60]. Considering the large variability of exercise capacity and strength levels reported by the current literature, clinicians should always include an appropriate functional evaluation with which to create a tailored exercise prescription in women with PCOS [61].

## Figures and Tables

**Figure 1 biomedicines-10-02176-f001:**
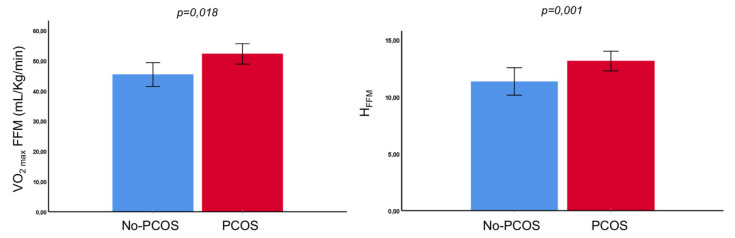
Differences in cardiorespiratory fitness, expressed as VO_2_maxFFM, and strength, expressed as H_FFM_, in PCOS patients and controls. Both values are statistically significantly higher in the PCOS group.

**Figure 2 biomedicines-10-02176-f002:**
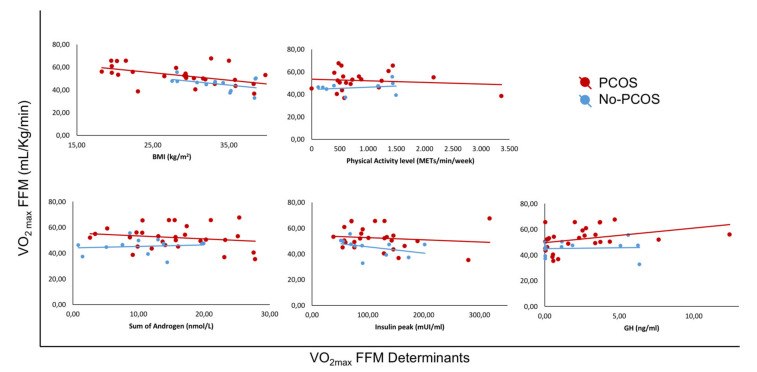
Linear correlations in various groups of patients (PCOS and controls) between cardiorespiratory fitness, expressed as VO_2_maxFFM, and its determinants: body mass index (BMI), physical activity level, androgenic pattern (sum of androgen), insulin peak and growth hormone (GH).

**Figure 3 biomedicines-10-02176-f003:**
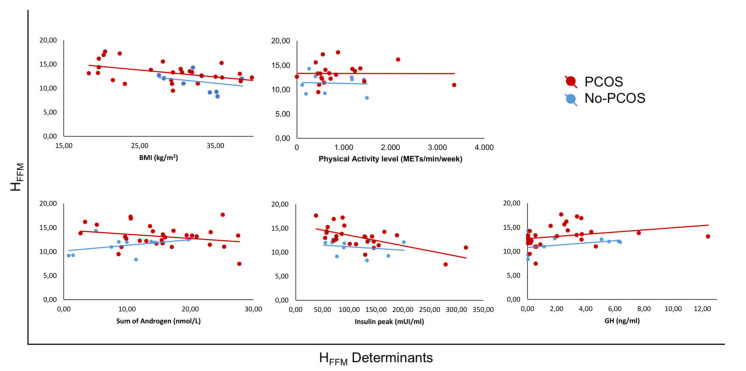
Linear correlations in different groups of patients (PCOS and controls) between strength level, expressed as H_FFM_, and its determinants: body mass index (BMI), physical activity level, androgenic pattern (sum of androgen), insulin peak and growth hormone (GH).

**Table 1 biomedicines-10-02176-t001:** Anthropometric, hormonal and body composition characteristics of PCOS patients and controls.

Characteristics	PCOS	No PCOS	*p*
	(n = 31)	(n = 13)	
Age (yr)	27.1 ± 4.8	30.6 ± 9.5	0.131
Weight (kg)	79.70 ± 18.51	90.06 ± 10.54	0.070
Height (m)	1.65 ± 0.05	1.65 ± 0.06	0.748
BMI (kg/m^2^)	30.3 ± 6.6	33.0 ± 4.2	0.138
Physical Activity Level (METs/min week)	556 ± 271	783 ± 651	0.090
Testosterone (nmol/L)	1.94 ± 1.11	1.61 ± 1.27	0.475
Androstenedione (nmol/L)	10.88 ± 5.10	5.40 ± 2.26	0.218
17-OH-P (nmol/L)	2.39 ± 1.67	1.71 ± 1.26	0.501
SHBG (nmol/L)	33.55 ± 15.65	52.33 ± 42.16	0.460
Free Androgen Index	7.27 ± 5.21	3.37 ± 3.25	0.079
Sum of Androgen (nmol/L)	14.72 ± 6.96	5.79 ± 5.36	0.274
Growth Hormone (ng/mL)	2.44 ± 2.78	1.32 ± 2.52	0.956
Insulin Peak (mUI/L)	118.16 ± 64.39	101.08 ± 49.08	0.369
Insulin Sensitivity (%) *	76.7	53.8	0.135
HOMA-ir	2.02 ± 1.40	2.70 ± 1.54	0.186
Fat Mass (kg)	30.18 ± 11.71	38.27 ± 7.17	0.078
Lean Mass (kg)	46.58 ± 8.25	48.50 ± 5.00	0.342
Lean Mass Percentage (%)	64.98 ± 9.28	54.58 ± 2.84	0.060
Lean Mass Dominant Arm (kg)	21.72 ± 5.17	21.52 ± 2.26	0.544

Data are expressed as means ± standard deviation. PCOS: polycystic ovary syndrome; BMI: body mass index; 17-OH-P: 17-OH-progesterone; SHBG: sex hormone binding protein; HOMA: homeostasis model assessment. * percentage of positivity to insulin test.

**Table 2 biomedicines-10-02176-t002:** Cardiorespiratory, functional and strength parameters of PCOS patients and controls.

Characteristics	PCOS	No PCOS	*p*
	(n = 31)	(n = 13)	
VO_2max_ (mL/min)	2424.3 ± 304.2	2216.5 ± 352.0	0.060
**OUES (mL/min∙logL)**	2213.5 ± 346.9	1937.2 ± 387.2	** *0.043* **
**VO_2_maxkg (mL/kg/min)**	30.9 ± 7.6	24.8 ± 4.1	** *0.010* **
**VO_2_maxFFM (mL/kg/min)**	52.4 ± 8.9	45.3 ± 6.2	** *0.018* **
**Exercise capacity (METs)**	14.4 ± 2.72	12.5 ± 1.72	** *0.019* **
Handgrip (kg)	28.27 ± 4.33	26.13 ± 5.4	0.052
**H_kg_**	0.36 ± 0.09	0.30 ± 0.08	** *0.009* **
**H_FFM_**	13.03 ± 2.32	11.50 ± 1.91	** *0.001* **

Data are expressed as means ± standard deviation. PCOS: polycystic ovary syndrome; VO_2_max, maximal aerobic capacity; OUES, oxygen uptake efficiency slope; VO_2_maxkg, cardiorespiratory fitness max per kg; VO_2_maxFFM, cardiorespiratory fitness peak per kg fat-free mass; H_Kg_, handgrip per kg; H_FFM_, handgrip per kg arm fat-free mass.

## Data Availability

Not applicable.

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
