# Peer review of "Do Androgenic Pattern, Insulin State and Growth Hormone Affect Cardiorespiratory Fitness and Strength in Young Women with PCOS?"

_biomedicines, 2022, doi:10.3390/biomedicines10092176_

Round 1

Reviewer 1 Report

Dear Authors my comments,

1. Introduction lines from 61-67; started from "Kogure" should be in discussion in my opinion.

2.English corrections.

3. Size of groups are limitations

4. Too old references.

5. Too many details in Materials and Methods part in my opinion

Author Response

Revision of the manuscript “Do androgenic pattern, insulin state and growth hormone affect Cardiorespiratory Fitness and Strength in young women with PCOS?”

We would like to thank the Editorial team and the Reviewers for the thorough review that our manuscript has received and for having appreciated our research work adding useful suggestions to further improve our manuscript. We hope that after our accurate revision you will find it suitable for publication in your prestigious Journal.

Point-by-point response:

  1. Introduction lines from 61-67; started from "Kogure" should be in discussion in my opinion.

Thanks for this suggestion, we reconsidered the paragraph and modified as follows since the topic has already been discussed in detail in the discussion “Conflicting results were found in the literature when assessing strength in women with PCOS [17, 18].”

  1. English corrections

Before submitting the paper, it was checked by a native English reviewer. Nevertheless, we adjusted some other points at this stage revision (lines16, 40, 66,188,395, 494).

  1. Size of groups are limitations

This issue was added in our limitation paragraph, line 487 “However, there are some limitations as well. The size of groups is small, and the subjects analysed were with obesity, probably due to the random sample selection.

  1. Too old references.

We added new ones:

2 - Chen W, Pang Y. Metabolic Syndrome and PCOS: Pathogenesis and the Role of Metabolites. Metabolites. 2021 Dec 14;11(12):869. doi: 10.3390/metabo11120869. PMID: 34940628; PMCID: PMC8709086.

17- Kogure GS, Ribeiro VB, Gennaro F, Ganoa de Oliveira F ,Rui A et al. Physical Performance Regarding Handgrip Strength in Women with Polycystic Ovary Syndrome. Rev Bras Ginecol Obstet 2020; 42, 12; 811-819, 21.12.2020- 0100-7203. DOI - 10.1055/s-0040-1718953.

63 - Woodward A, Klonizakis M, Broom D. Exercise and Polycystic Ovary Syndrome. Adv Exp Med Biol. 2020;1228:123-136. doi: 10.1007/978-981-15-1792-1_8. PMID: 32342454.

  1. Too many details in Materials and Methods part in my opinion

We cut out the really detailed part of material and methods related to endocrinological laboratory medicine, DEXA and cardiopulmonary exercise testing a follows:

Line 85 . “All the partecipants underwent to a three-hour oral glucose tolerance test (OGTT), maintaining an intake of at least 300g of glucose in the two days before the test. Normal weight was considered with BMI between 18.0 and 24.9 (kg/m2) or obese at a BMI higher than 30, as well as being considered normoinsulinemic (maximum insulin peak ≤60 mIU/L, ISI≥ 6, and HOMA-ir < 1.8) or hyperinsulinemic (maximum insulin peak > 70 mIU/L, ISI < 4, and HOMA-ir ≥ 2.5 [24].”

Line 193: “All blood samples were obtained during the early follicular phase (day 2–5) of a spontaneous or progesterone-induced menstrual cycle. Glucose was measured by means of the glucose oxidase method (Gluco-quant®; Roche Diagnostics GmbH, Mannheim, Germany). LH, FSH and estradiol were assessed by a competitive immunoassay with the use of an electrochemiluminescence immunoassay (ECLIA; Roche Diagnostics GmbH, Mannheim, Germany). GH was measured by chemiluminescent immunometric assay (Medical System S.p.a., Genova, Italy)[25]. Serum Testosterone and DHEA-s were further measured by competitive immunometric chemiluminescent enzyme immunoassay (Immulite 2000; Diagnostic Products Corporation, Los Angeles, USA). Androstenedione and 17-OH-P were measured by enzyme immunoassay (ELISA) (DRG Instruments GmbH, Marburg, Germany).  SHBG was measured by two-site chemiluminescent immunometric assay (Immulite 2000; Diagnostic Products Corporation, Los Angeles, USA). Cholesterol and triglycerides were..”

Line 209: “and electrocardiogram monitoring during the entire test (Cardiosoft EKG System, GE, USA). Blood pressure was measured by a trained medical doctor using the auscultatory method. Breath-by-breath ventilation was measured with a low-resistance turbine to determine inspiratory and expiratory volumes and flow during the exercise test. Inspired and expired gases were sampled continuously at the mouth and analysed for concentrations of O2 and CO2 via mass spectrometry after calibration with analysed gas mixtures. The treadmill test consisted of a maximal Bruce Ramp protocol, characterised by progressive speed and inclination increments. The interruption criterion was the achievement of a Rate of Perceived Exertion ≥ of 18/20 on the Borg Scale [26], which is associated with a heart rate (HR) value ≥ 85% of the age-predicted maximum HR (220 bpm – age) and/or with a respiratory exchange ratio (RER) value > 1.10. A treadmill was preferred to a cycle-ergometer because of its more reliable value for VO2max, which is not limited by peripheral muscle fatigue [27, 28]. Body composition was evaluated via Dual X-ray Absorptiometry (DXA) (model DXA QDR 4500 W, software Version 12.6; Hologic, Bedford, MA, USA). This method followed the International Society for Clinical Densitometry Guidelines [29]. Handgrip dynamometer was used to evaluate the dominant and non-dominant upper limb strength (Baseline, Elmsford, NY, US). The fit of the hand on the grip was adjusted to accommodate the right size of the participant. The elbow was flexed to 90° to guarantee the strongest grip strength measurement [30]. The results were evaluated in percentile form with respect to the reference population and as raw data [13]. The physical activity level was assessed with the Global Physical Activity Questionnaire (GPAQ). Women were classified either as “active” or “sedentary” following the World Health Organization (WHO) recommendations for weekly levels of physical activity [31]. A validated Italian version of the Questionnaire was administered by a trained kinesiologist to ensure data reliability [32].

Reviewer 2 Report

The authors focused on the study of Do androgenic pattern, insulin state and growth hormone affect Cardiorespiratory Fitness and Strength in young women with PCOS. This is an interesting and comprehensive article. The article is well structured.

In my opinion:

- The abstract presents an accurate description of this study.

- An Authors was conducted adequate literature review

- References support the rationale for reporting the study.

- Patients are described adequately.

- The management of the study is effectively described.

- Valid and reliable outcome measures are utilized.

- Conclusions are appropriate.

Key points to consider:

-        In the article sometimes the names of the Authors are spelled differently, please correct this with one of the versions (for example Line 61 and 63).

-        Kg or kg - please correct this to one of the versions throughout the article.

Overall impression about the quality of the study is good.

Author Response

Revision of the manuscript “Do androgenic pattern, insulin state and growth hormone affect Cardiorespiratory Fitness and Strength in young women with PCOS?”

We would like to thank the Editorial team and the Reviewer for the thorough review that our manuscript has received and for having appreciated our work, adding useful suggestions to further improve our manuscript. We hope that after our accurate revision you will find it suitable for publication in your prestigious Journal.

Point-by-point response:

The authors focused on the study of Do androgenic pattern, insulin state and growth hormone affect Cardiorespiratory Fitness and Strength in young women with PCOS. This is an interesting and comprehensive article. The article is well structured.

In my opinion:

- The abstract presents an accurate description of this study.

- An Authors was conducted adequate literature review

- References support the rationale for reporting the study.

- Patients are described adequately.

- The management of the study is effectively described.

- Valid and reliable outcome measures are utilized.

- Conclusions are appropriate.

Key points to consider:

  1. In the article sometimes the names of the Authors are spelled differently, please correct this with one of the versions (for example Line 61 and 63).

We completely modified the sentence, solving as follow the spelling problem: “conflicting results were found in the literature when assessing strength in women with PCOS [17, 18].

  1. Kg or kg - please correct this to one of the versions throughout the article.

It has been modified as suggested: from Kg to kg.

Overall impression about the quality of the study is good.

Thanks a lot for your precious suggestions

Round 2

Reviewer 1 Report

Dear Authors,

I accept your response.